# Composite Fe-Cr-V-C Coatings Prepared by Plasma Transferred-Arc Powder Surfacing

**DOI:** 10.3390/ma16145059

**Published:** 2023-07-18

**Authors:** Xin Zhang, Yong Liu, Huichao Cheng, Kun Li, Cheng Qian, Wei Li

**Affiliations:** State Key Laboratory of Powder Metallurgy, Central South University, Changsha 410083, China; zxx3482@163.com (X.Z.); yonliu@csu.edu.cn (Y.L.); kunlee@csu.edu.cn (K.L.); candyq94@163.com (C.Q.)

**Keywords:** plasma transferred-arc powder surfacing, iron-based composite powder, arc current, ion gas flow rate

## Abstract

In this study, we developed composite Fe-Cr-V-C coatings by plasma transferred-arc (PTA) powder surfacing on a 42CrMo steel substrate. The effects of arc current and ion gas flow rate on the coatings’ microstructure, hardness, and bonding performance were investigated. During the surfacing process, $V_{x}C_{y},{M_{7}C}_{3}\left(M=Fe,Cr\right)$ and other hard phases are in-situ generated throughout the entire PTA powder surfacing. These phases are uniformly distributed in the Fe matrix through precipitation and dispersion strengthening, yielding a surface hardness of up to 64.1 HRC. Also, the bonding performance between the substrate and coatings was evaluated by measuring the tensile strength, revealing that strong metallurgical bonds are formed, reaching a strength greater than 811 MPa.

## 1. Introduction

The rapid development of modern industry has inevitably exposed mechanical parts to more complex and demanding working environments and conditions [1,2,3]. As a medium- and low-carbon alloy structural steel with good overall mechanical properties, 42CrMo steel is widely used in the machinery manufacturing industry, particularly for tunnelling excavation and mining excavation machines [4], such as the pick magazine body for cantilever tunneling machines and the ball tooth magazine body for down-the-hole drilling machines. Due to hard minerals, the pick undergoes high cyclic compressive stress, shear stress, and impact load during the mining process. This causes problems such as alloy head fragmentation and tooth body wear, increasing the wear rate of the pick [5,6]. Under such complex working conditions, cutting picks must have good wear resistance and impact performance.

Currently, the common surface treatment methods of 42CrMo steel primarily include nitriding, shot peening, quenching, coating technology, and others [7,8]. The research on using plasma transferred-arc (PTA) powder surfacing to improve overall workpiece performance has recently been extensively developed [9,10]. Compared with laser cladding, plasma powder surfacing is a simple, highly-efficient, and cost-effective process, yielding strong bonding between the prepared coating and the selected substrate [11]. In addition, since most alloys can be processed as powders, the plasma powder surfacing process can be used with a variety of overlay powders and is not limited by some material’s properties that tend to affect processing (e.g., ductility), making the process widely used in manufacturing applications, such as for mining cutoffs [12].

PTA powder surfacing mostly uses Ni-based, Co-based, and other self-melting metal alloy or composite powders with high particle sphericity and good fluidity as raw materials. Wei et al. [13] prepared a surface layer made of the Ni + WC composite powder on the surface of 45 steel by plasma powder surfacing and studied the effect of the surfacing process on its properties. When the welding current was 120 A, an overheating effect occurred, leading to severe melting of WC and the formation of Fe_3_W_3_C, FeW_3_C, and Ni_2_W_4_C phases, which decreased the hardness of the welding layer. Appiah et al. [14] mixed Ni-Si-B + 60 wt.% WC and Ni-Cr-Si-B + 45 wt.% WC metal matrix composite (MMC) powders and applied plasma powder surfacing technology to prepare a surfacing coating on AR400 steel. A dendritic structure composed of primary WC particles and secondary WC precipitates was formed in the overlay layer, which improved the wear resistance of the overlay layer by 5.7 times compared to the substrate.

The current research focus on PTA surfacing of powder raw materials is mainly on the composite design of self-fluxing alloy powders and related reinforced powders. Although some achievements were made, new research areas have not significantly expanded [15]. The composition pallet of surfacing powder raw materials is relatively small, and there is a lot of research on nickel-based and cobalt-based self-fluxing alloy powders, showing good application development. However, research on iron-based and copper-based composite powder systems and high-entropy alloy powder materials is limited. In addition, in the self-fluxing alloy powder system, reinforcing is mainly achieved via adding WC and VC particles, and there is practically only one reinforcement method, which hardly meets the growing technical needs in different fields. In response to the above issues, it is necessary to continuously innovate the design ideas of materials in the powder system and develop new types of surfacing powders, such as metal-ceramic composite powders, amorphous powders, and in-situ reaction powders [16]. From the perspective of powder preparation technology, it is necessary to address the issues of uneven powder composition and the formation of holes [12].

Compared with Ni, Co, and other limited resources, Fe-based powder is widely available, affordable, and capable of dissolving carbides to form a eutectic [17,18]. The organization and functionality of the coating can be improved through solid solution strengthening, dispersion strengthening, fine grain strengthening, etc., via adding Cr, V, Si, Mo, and other strengthening elements to Fe powder [19]. This study uses the PTA powder surfacing technology to prepare Fe-Cr-V-C coatings on 42CrMo steel. By adding elements such as Cr and V to the powder raw material, hard phases such as M_7_C_3_ (M=Fe,Cr) and VC are formed in-situ during the surfacing process. The excellent properties of high melting point and high hardness carbides are utilized to improve the hardness and strength of the surface coating. We varied the arc current and ion gas flow rate process parameters and used the tensile strength as a parameter to characterize the bonding properties between the substrate and the coatings, and finally optimized the process parameters to obtain a favorable result.

## 2. Experimental Part

### 2.1. Materials Preparation

The substrate was a 42CrMo steel plate material (80 mm × 34 mm × 37 mm). To determine the bonding strength between the substrate and the coating through the subsequent Tensile testing, that is, it is necessary to ensure that the parallel sections of the tensile specimen are all coatings, so a 34 mm × 5 mm × 5 mm groove was cut at each end of the plane as a welded channel by wire cutting (Figure 1).

The powder used in the investigation was X-FeCrV15 iron-based alloy powder (gas atomization preparation) with a particle size of 75–130 μm. The chemical composition of matrix 42CrMo steel is shown in Table 1. The chemical composition of the iron-based powder is listed in Table 2.

### 2.2. PTA Powder Surfacing Process

The PTA powder surfacing test was carried out using a DML-V03BD plasma powder surfacing machine (Duomu Industrial Co., Ltd, Shanghai, China), with high-purity argon as the shielding gas, and the process parameters are shown in Table 3.

### 2.3. Microanalysis

We used an X-ray diffraction analyzer (Smartlab SE, Rigaku, Tokyo, Japan) for the phase comparison analysis of the surfacing powder and layer (diffraction angle range 30° < 2θ < 80°, rate 5°/min). The fracture morphology, surface and cross-sectional morphology of the deposited layer, and changes in the micro-zone composition after stretching were analyzed using a TESCAN MIRA scanning electron microscope (TESCAN, Brno, Czech Republic).

### 2.4. Tensile Strength and Hardness Tests

Tensile tests were performed using an INSTRON3369 (INSTRON, USA) electronic universal material testing machine (the size of the tensile sample is shown in Figure 2, the thickness is 1.5 mm) to obtain the tensile strength of the welded specimen. The hardness (HV) was measured using an HVS-5 Vickers micro-hardness tester (BUEHLER, Lake Bluff, IL, USA); the coatings were analyzed at 10 points, with an increment of 0.65 mm, while the substrate was tested at 2 points with the same increment to measure the hardness of the coatings’ cross-section; the test load was 0.5 kg, and the holding time was 155 s. A Rockwell hardness tester (WH2002T, Wilson, KS, USA) was used to perform hardness testing on the surface of the deposited layer, and the average value of hardness (HRC) was determined from 3 measurement points.

## 3. Results and Discussions

### 3.1. Microstructural Characterization

Figure 3a–f show the SEM images of the Fe-Cr-V-C powder used for PTA powder surfacing. Mostly, the powder exhibits spherical or near-spherical particles, which is favorable for reducing the cohesion and friction between the particles. Moreover, there are no fragmented or satellite particles, which improves the powder’s fluidity. However, some pores can be observed on the powder’s surface, which may contribute to its low space-filling efficiency.

According to the EDS-determined elemental content in Figure 3f and Table 4, combined with the XRD data in Figure 4, the dispersion distribution of the black hard phase A in the Fe-based powder is V-rich carbides, such as VC and $V_{6}C_{5}$, and the layered dark gray phase B is the Fe phase, and there are some Cr-rich carbides M_7_C_3_ (M=Fe,Cr). The white phase C is distributed around the black phase, and its composition is similar to that of the dark gray phase B. The content of the Fe phase is relatively high.

Figure 4 shows the XRD patterns of the powder and surfacing coatings prepared under different parameters. Figure 5c and Table 5 show the SEM and EDS analysis of the characteristic areas of the coating surface prepared under an arc current of 110 A and an ion gas flow rate of 2.5 L/min. The analysis shows that the arc current and ion gas flow rate have little effect on the phase composition of the welding coating. The main crystalline phases of the surface coating are α-Fe, γ-Fe, V_x_C_y_ (VC,V_6_C_5_,VC_0_._845_,VC_0_._75_), and M_7_C_3_ (M=Fe,Cr) [20]. When the arc current and ion gas flow rate increase, the intensity of diffraction peaks increases.

The intensity of diffraction peaks of Fe-based coating crystals prepared at an arc current of 120 A and an ion gas flow rate of 2.5 L/min is large, and the γ-Fe phase is obvious, which may be related to the crystallinity of the crystals in the surface coating [21,22]. When the arc current and gas flow rate increase, the alloy powder fully melts, and the VC- and MC-type eutectic crystals remain at high temperatures for a longer time, exhibiting a higher crystallinity than under other processes [23]. Therefore, the peak intensity in the XRD pattern is relatively large.

Based on the XRD pattern of the surface coatings in Figure 4, the SEM image of the surface coating in Figure 5c, and the EDS elemental content analysis of the surface coating in Table 5, it can be seen that the surface coating is divided into three regions: spherical black particles (A), dark gray dendritic and inter-dendritic eutectic (B), and light gray, blocky features (C) [24], which are vanadium carbides, eutectic carbides of Cr and Fe, and matrix grains, respectively.

Figure 5 and Figure 6 show the SEM images of coating surfaces prepared under different arc currents and ion gas flow rates. A large amount of spherical V_x_C_y_ is dispersed in the surface coating substrate, and inter-dendritic M_7_C_3_ eutectic carbides are distributed at the grain boundaries in the intermittent network [25]. The reason for the reticulation of the hard phase of M_7_C_3_ is that the content of Cr is relatively low, so it cannot directly precipitate the M_7_C_3_ phase. It can only precipitate at the grain boundaries after the Fe phase is first precipitated and when the Cr concentration reaches a supersaturation level in the remaining liquid. With the arc current and gas flow rate increase, the V_x_C_y_ content increases, size decreases, and the dispersion distribution become more uniform while the number of dendrites decreases [26].

**Figure 5 materials-16-05059-f005:**
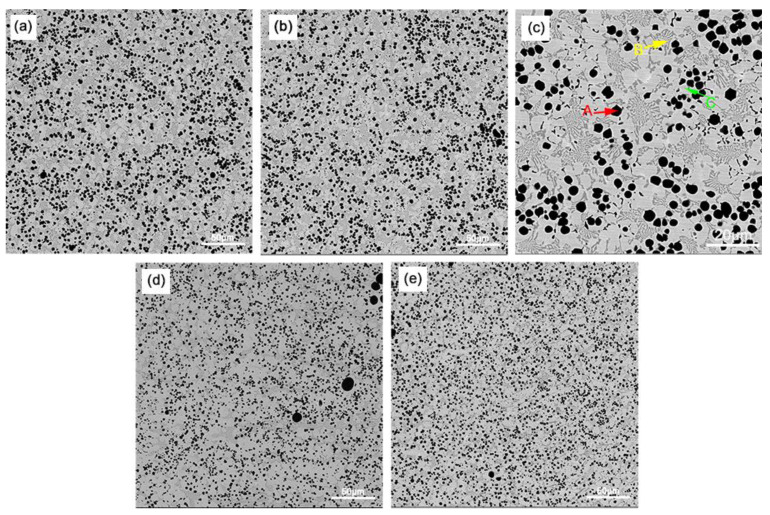
SEM images of the coating surface prepared under different arc currents (the ion gas flow rate is 2.5 L/min): (**a**) 100 A; (**b**) 110 A; (**c**) is hows the point of EDS scanning under 110 A; (**d**) 120 A; (**e**) 130 A.

**Table 5 materials-16-05059-t005:** EDS elemental content on the surface of the surface coatings (the arc current is 110 A, and the ion gas flow rate is 2.5 L/min).

	Black A	Dark Gray B	Light Gray C
Element	wt.%	wt.%	wt.%
C	13.18	10.09	4.28
V	74.31	10.03	4.22
Cr	7.11	27.42	9.98
Mn	0.09	1.15	1.08
Fe	2.72	48.32	77.51
Si	0.82	1.14	1.20
Ni	0.25	0.18	0.28
Mo	1.52	1.67	1.45

**Figure 6 materials-16-05059-f006:**
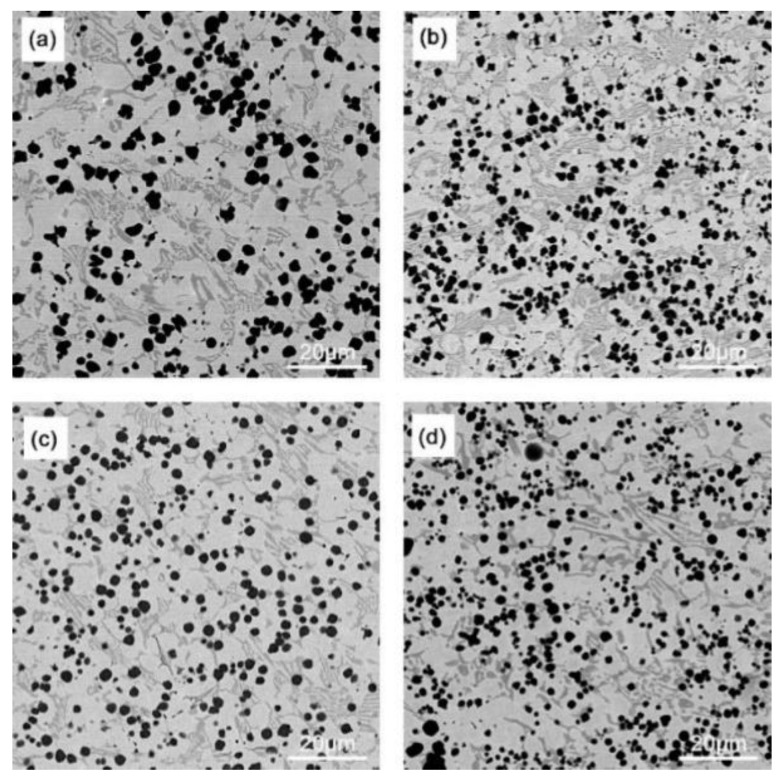
SEM images of the coating surface prepared under different ion gas flow rates (the arc current is 120 A): (**a**) 1.5 L/min; (**b**) 2.0 L/min; (**c**) 2.5 L/min; (**d**) 3.0 L/min.

Figure 7 shows the EDS mapping of the coating (the arc current is 120 A, and the ion gas flow rate is 1.5 L/min). The distribution of various elements in the coating is uniform, indicating that the surface layer has a good dilution ratio, metallurgical solid bonding with the substrate, and relatively uniform performance [27].

Under the thermal cycling effect of overlay welding, the solid substrate will generate a heat-affected zone. During the surfacing process, the alloy powder is melted into droplets, the reaction time between each element is short, and the reaction degree is intense, resulting in the formation of carbides between carbon elements and other elements in the alloy powder. Due to the convective mass transfer between the raw material powder and the matrix, the microstructure and appropriate binding of the matrix surface will change as the powder gradually cools and solidifies. In SEM images, different organizational structures can be observed from the interior of the surface layer, which is called the interface layer area (as shown by the red boundary in Figure 8). Figure 8 shows the morphology of the fusion zone under different arc currents, and Table 6 shows the width values of the fusion zone interface in Figure 8. Table 6 shows that due to the high pool temperature of PTA powder surfacing, which is conducive to gas release, there are no defects, such as pores and cracks near the fusion zone. Moreover, due to the small heat-affected zone of this process, the width of the fusion zone is relatively narrow. In addition, as the arc current increases, the energy of the plasma arc increases, the melting and penetration ability increases, the cooling and solidification time of the melt pool increases, and the convective mass transport between the substrate and the coating enhances, leading to a gradual increase in the width of the interface.

Figure 9 shows the EDS scanning point of the interface is at 110 A, 2.5 L/min. Table 7 shows the EDS-determined elemental analysis of the fusion zone in the surface coating. Figure 9 and Table 7 show that the elements of the matrix and coating diffuse each other, forming a hard phase in the fusion zone that is different from the inside of the surfacing layer. Combining Table 5, it can be found that compared to the inside of the coating, the content of Cr in the newly generated carbide hard phase at the interface is lower, while the content of V is higher (point B), which ensures the bonding strength between the matrix and coating. In addition, in the area near the interface, the V and Cr contents of rich V carbides (point A) and rich Cr carbides (point C) are higher. This may be caused by the deposition and segregation of V and Cr during the surfacing process.

Figure 10 shows the SEM images of the surface layer interface under two different surfacing processes. According to Figure 10a, it can be observed that there are planar crystals in the fusion zone. This is because the latent heat of crystallization released during the solidification of the surfacing powder near the substrate is released through the substrate. At this time, the undercooling of the component is approximately zero, and crystallization occurs under a positive temperature gradient, causing the crystals here to move forward in a nearly planar manner. With the surfacing process, the substrate temperature increases, and the cooling rate decreases. Under the condition of composition supercooling, the latent heat of crystallization can be released through the solid/liquid phase, thus forming columnar crystals (V_x_C_y_), dendrites and lamellar eutectic (M_7_C_3_).

From the line scanning (Figure 10b) and the mapping scanning (Figure 11) results near the fusion zone, it can be seen that elements have mutual diffusion between the substrate and the coating [28], indicating the formation of good metallurgical bonding during the plasma transferred-arc powder surfacing.

### 3.2. Tensile Strength

We used tensile strength to characterize the bonding strength between the substrate and the surface layer. Figure 12 and Figure 13 show the tensile strength of the sample after plasma surfacing as a function of the arc current and ion gas flow rate. As shown in Figure 12, as the arc current increases, the tensile strength first increases and then decreases. When the welding current is 120 A, the maximum tensile strength is 694 MPa. From Figure 5, when the welding current increases to 120 A, the content of the V_x_C_y_ hard phase increases and suppresses grain growth, achieving a dispersion-strengthening effect and the highest strength [28]. When the current increases to 130 A, the excessive hard phase content increases the brittleness and decreases the strength of the weld overlay. As shown in Figure 13, as the ion gas flow rate increases, the tensile strength first decreases and then increases. When the ion gas flow rate was 1.5 L/min, the maximum sample strength was 811 MPa, which may be because the ion gas flow rate at this point matched the protective gas flow rate and the arc current, allowing the arc to reach stability [29]. As the ion gas flow rate increases, the arc stiffness increases, the molten pool temperature formed during plasma surfacing is higher, and the carbide content in the coating increases (as shown in Figure 6), resulting in a decrease in tensile strength.

Figure 14 shows the macroscopic appearance of the tensile fracture sample prepared under an arc current of 100 A and an ion gas flow rate of 2.5 L/min, Figure 14a, and an arc current of 120 A and an ion gas flow rate of 1.5 L/min, Figure 14b. The tensile fracture surfaces of the samples with the highest and lowest strength are located in the middle of the welding layer, and the connection region between the welding layer and the substrate is well bonded, without any fracture. The macroscopic fracture in Figure 14 shows no significant elongation, bending, or necking, and in the fracture morphology under three different surfacing processes shown in Figure 15, obvious cracks and river-like patterns propagate towards the grain. Determine whether the tensile fracture mode of the overlay layer is a brittle fracture. This is due to forming a network of brittle phases (M_7_C_3_) during the surfacing process. Therefore, under external forces, these network brittle phases will directly bear the load, which is easy to break and form cracks, causing cracks to propagate along grain boundaries, resulting in intergranular fracture of the surfacing layer.

### 3.3. Hardness

Figure 16 and Figure 17 show the variation curves of the surface hardness with arc current and ion gas flow rate, respectively. According to the analysis of Figure 16 and Figure 17, as the arc current and ion gas flow rate increase, the surface hardness of the welding layer first increases and then decreases. In Figure 16, when the arc current is between 100–120 A, the distribution of hard phases (V_x_C_y_, M_7_C_3_) in the coating is more uniform due to the full melting of the powder. Therefore, the surface hardness of the welding layer slightly increases with the arc current, but the change is not significant. Compared to the surface coatings prepared at 100 A–120 A, the surface hardness of the coating prepared at 130 A significantly decreases, plausibly due to the excessive heat input caused by the excessive arc current, which increases the dilution rate of the surfacing layer and changes the distribution of hard phases due to atomic diffusion, resulting in a decrease in the surface hardness of the surface layer when the arc current is 130 A [30].

Figure 17 shows that the plasma arc is stable when the ion gas flow rate is 1.5–2.5 L/min, with moderate stiffness and uniform phase distribution. Especially when the ion gas flow rate is 2.5 L/min, the surface hardness of the surface coatings is the highest, reaching 64.1 HRC, which is consistent with the uniform distribution of hard phases in Figure 6. When the ion gas flow rate increases to 3.0 L/min, the hardness of the surface layer decreases, which may be due to higher arc stiffness and melting depth, increasing the dilution rate and decreasing the hardness.

Figure 18 and Figure 19 show the cross-sectional hardness of surface coatings prepared under different currents and ion gas flow rates. Due to the in-situ generation of V_x_C_y_ and M_7_C_3_ hard phases during plasma surfacing, the hardness of the surface coating is significantly higher than that of the 42CrMo steel substrate. In Figure 18, the hardness at the fusion line of the surface layer prepared under 120 A is significantly lower. As shown in Figure 8b, the molten pool formed at 120 A is subjected to a higher temperature, facilitating the diffusion of the low-melting-point Fe phases, resulting in a wider fusion zone and an increase in the dilution rate, thereby reducing the interfacial hardness [27]. In Figure 19, there is no significant difference in the hardness of the fusion zone of the surface layer prepared under an ion gas flow rate of 2.0–3.0 L/min. When the ion gas flow rate is 1.5 L/min, the fusion interface hardness is the highest. This may be because, as the ion gas flow rate increases, the plasma arc stiffness and the melting depth increase, making the dilution faster and changing the distribution of the hard phase [31]. Therefore, the fusion interface hardness of the surface layer prepared at 1.5 L/min is significantly higher than that of the surface layer prepared at 2.0–3.0 L/min.

## 4. Conclusions

In this study, we investigated the Fe-Cr-V-C composite powder surface layers prepared by the PTA powder surfacing process with a welding current of 100–130 A and an ion gas flow rate of 1.5–3.0 L/min. The following conclusions can be drawn:

1The overlay layer is mainly composed of α-Fe, γ-Fe, V_x_C_y_ (VC,V_6_C_5_,VC_0_._845_,VC_0_._75_), M_7_C_3_ (M=Fe,Cr) and other phases, which can significantly improve the hardness of the surface coating.2The Fe-Cr-V-C plasma surface layer exhibits good metallurgical bonding with the substrate, and the tensile fractures do not occur at the junction. There are no fractures and a good connection between the welding layer and the substrate. The surface layer is free of defects, e.g., pores and slag inclusions.3We used tensile strength to characterize the bonding strength between the substrate and the surface layer. The tensile strength varies with different welding process parameters. As the arc current increases, the tensile strength and surface hardness first increase and then decrease. With the increase in the ion gas flow rate, the tensile strength first decreases and then increases, while the surface hardness first increases and then decreases.

## Figures and Tables

**Figure 1 materials-16-05059-f001:**
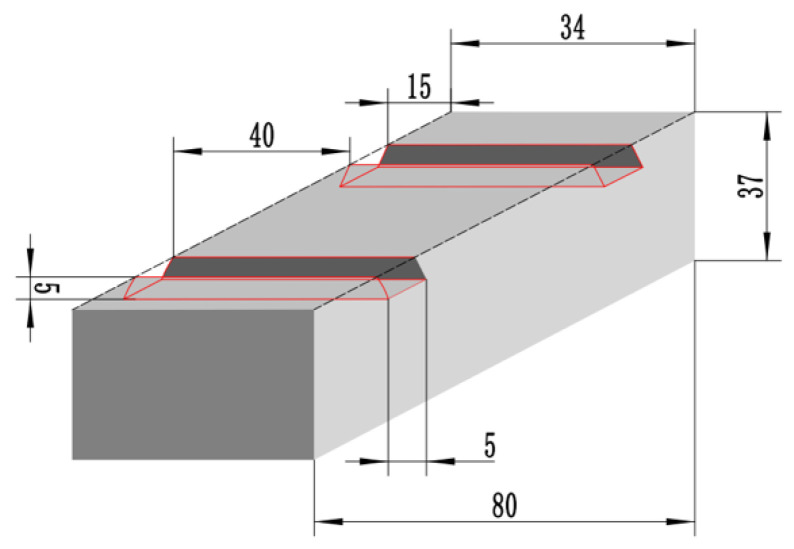
The diagram of the substrate weld channel size.

**Figure 2 materials-16-05059-f002:**
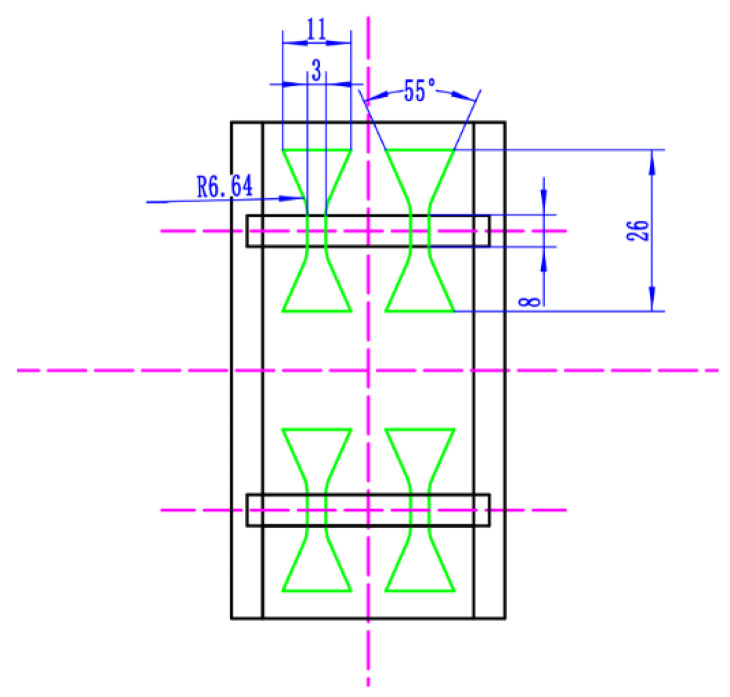
The schematic diagram of the cutting size of the tensile sample line.

**Figure 3 materials-16-05059-f003:**
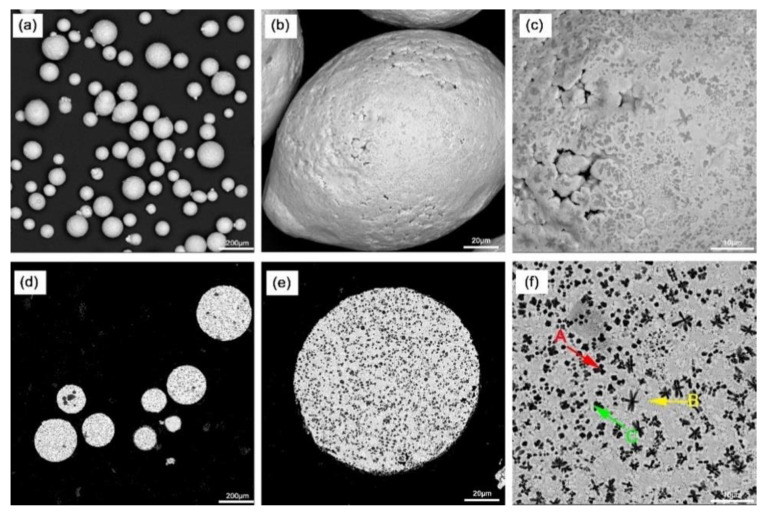
Surface and cross-sectional morphology of the Fe-based powder: (**a**–**c**) surface morphology, and (**d**–**f**) cross-sectional morphology.

**Figure 4 materials-16-05059-f004:**
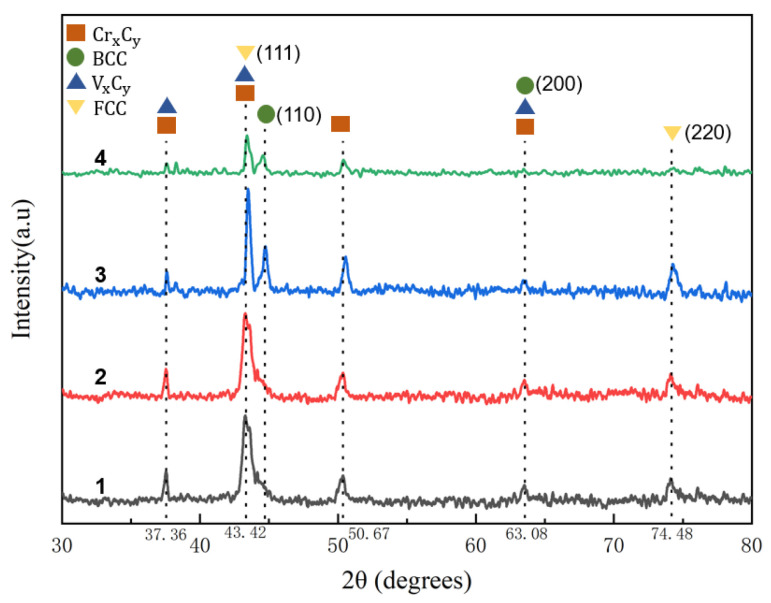
XRD patterns of the powder and surface coatings prepared at different process parameters: 1—Fe based powder; 2—110 A 2.5 L/min coatings; 3—120 A 2.5 L/min coatings; 4—120 A, 1.5 L/min coatings.

**Figure 7 materials-16-05059-f007:**
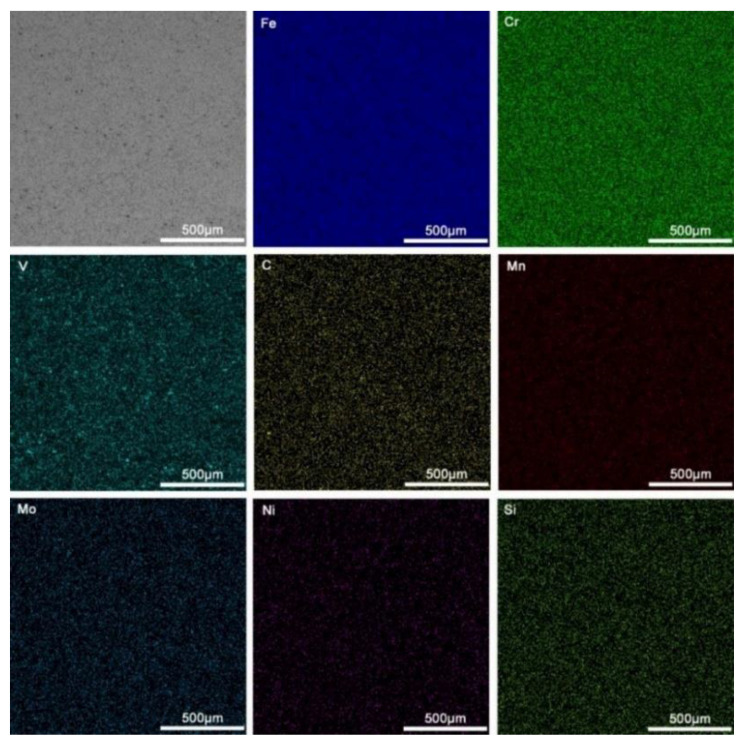
EDS surface mapping of the coating prepared at an arc current of 120 A and an ion gas flow rate of 1.5 L/min.

**Figure 8 materials-16-05059-f008:**
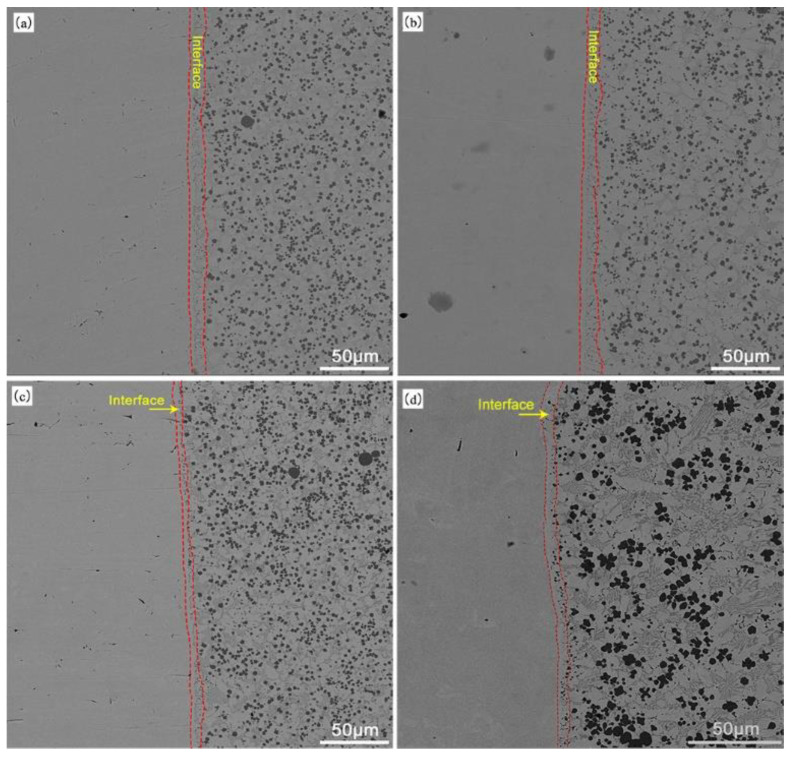
Width of fusion zone under different arc currents: (**a**) 130 A; (**b**) 120 A; (**c**) 110 A; (**d**) 100 A.

**Figure 9 materials-16-05059-f009:**
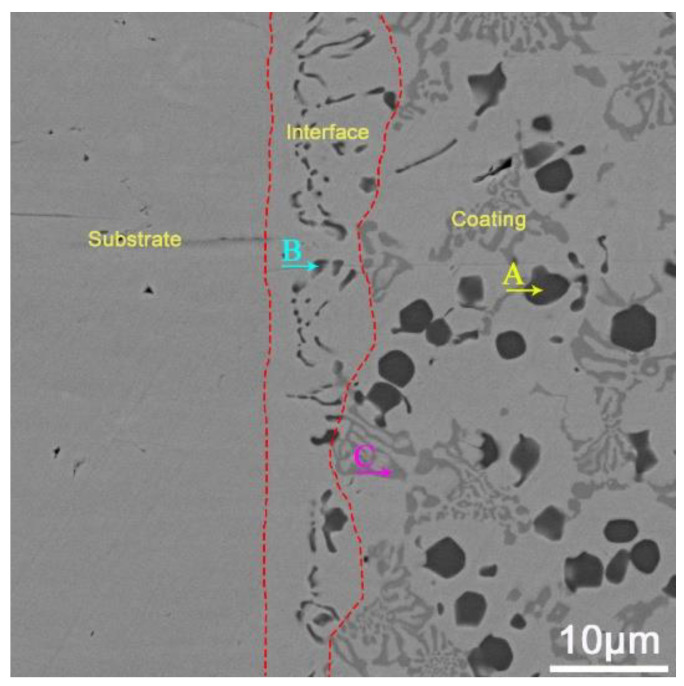
The EDS scanning point of the interface is at 110 A, 2.5 L/min.

**Figure 10 materials-16-05059-f010:**
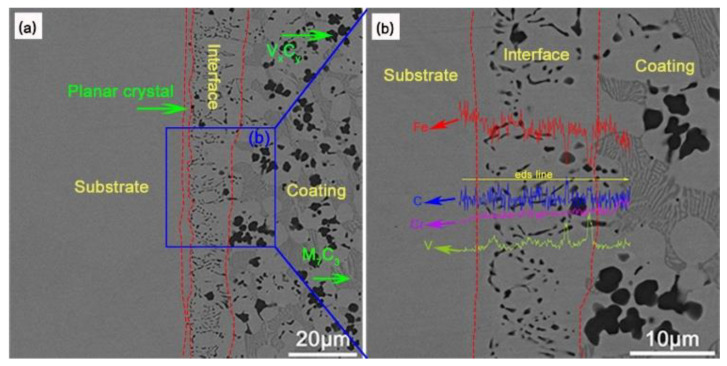
SEM images of the interface of the surface layer prepared under 120 A, 1.5 L/min: (**a**) 2.00 kx; (**b**) 5.00 kx.

**Figure 11 materials-16-05059-f011:**
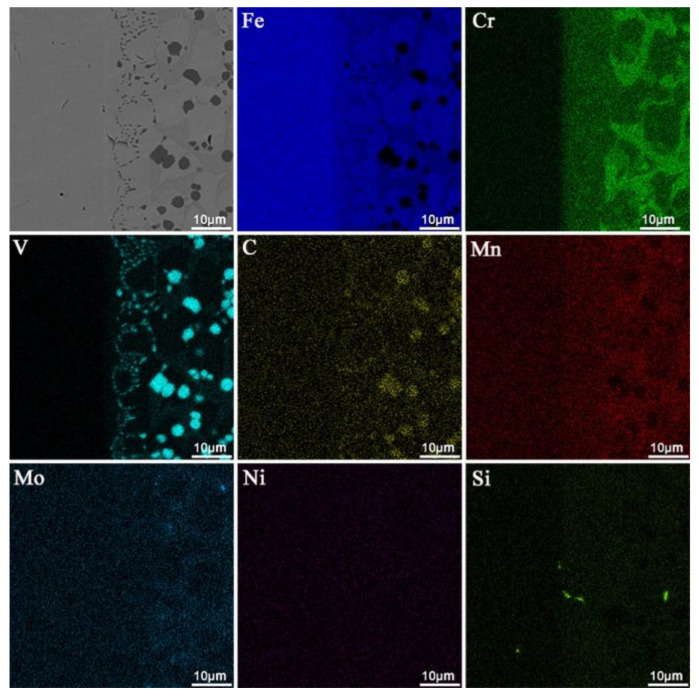
EDS interface mapping of the coating prepared at an arc current of 130 A and an ion gas flow rate of 2.5 L/min.

**Figure 12 materials-16-05059-f012:**
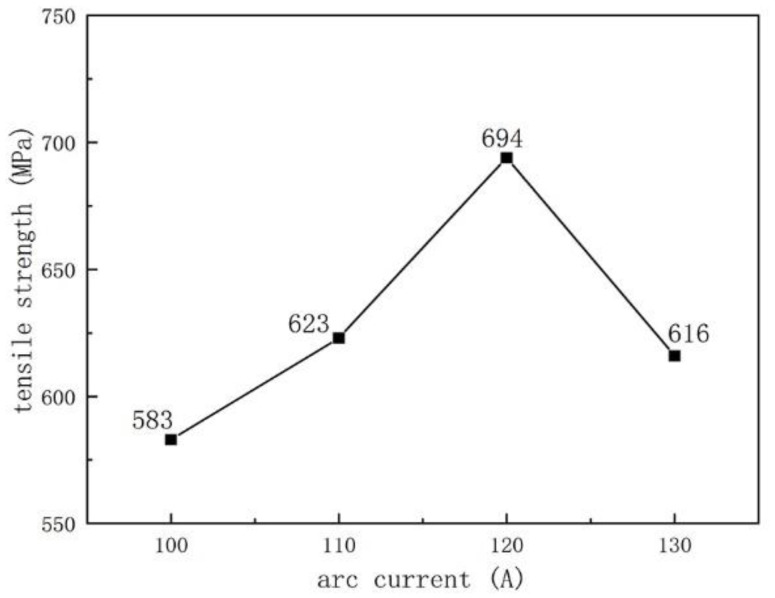
The change of the tensile strength with the arc current (the ion gas flow rate is 2.5 L/min).

**Figure 13 materials-16-05059-f013:**
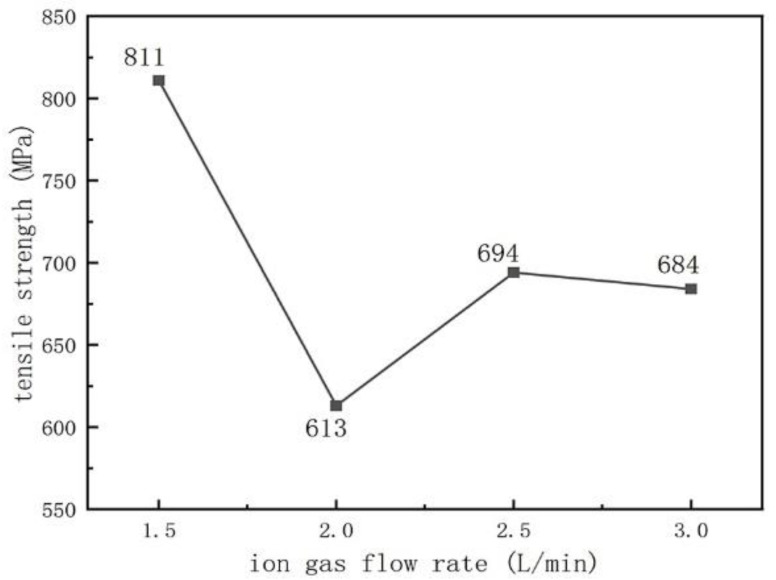
The change of the tensile strength with the ion gas flow rate (the arc current is 120 A).

**Figure 14 materials-16-05059-f014:**
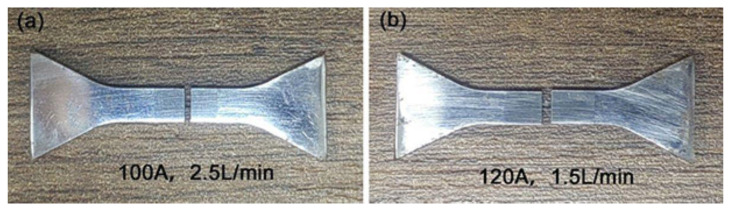
The tensile fracture samples were prepared under different process parameters: (**a**) 100 A, 2.5 L/min, and (**b**) 120 A, 1.5 L/min.

**Figure 15 materials-16-05059-f015:**
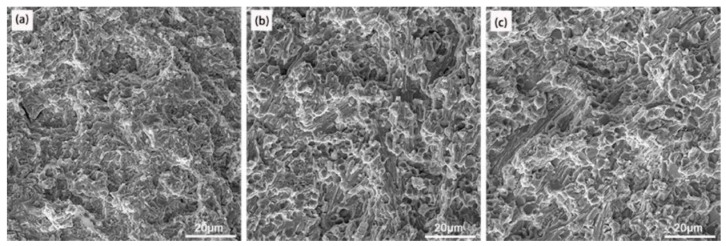
Fracture morphology: (**a**) 120 A, 2.5 L/min; (**b**) 130 A, 2.5 L/min; (**c**) 120 A, 1.5 L/min.

**Figure 16 materials-16-05059-f016:**
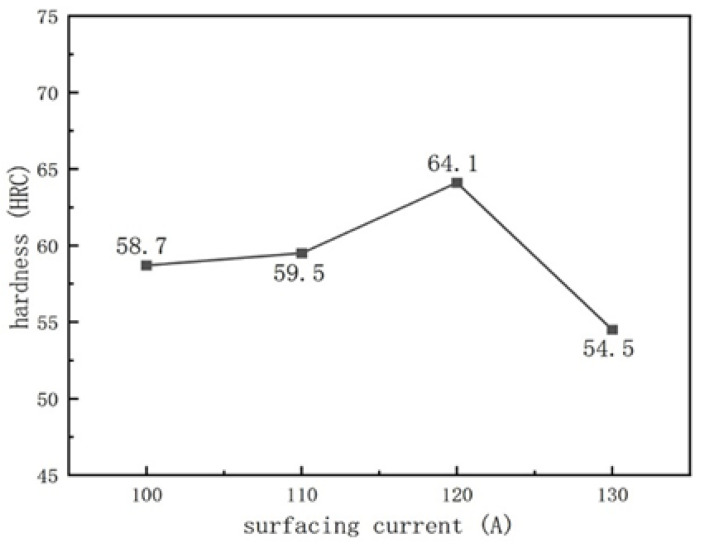
The change of the coating hardness with the arc current: (the ion gas flow rate is 2.5 L/min).

**Figure 17 materials-16-05059-f017:**
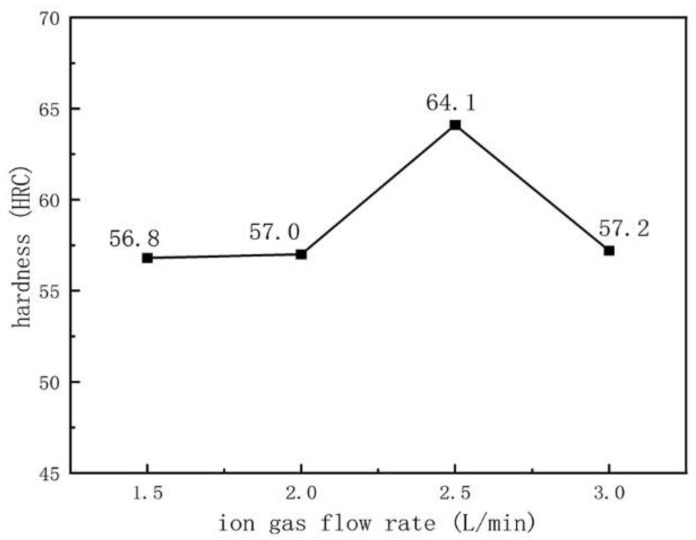
The change of the coating hardness with the ion gas flow rate: (the arc current is 120 A).

**Figure 18 materials-16-05059-f018:**
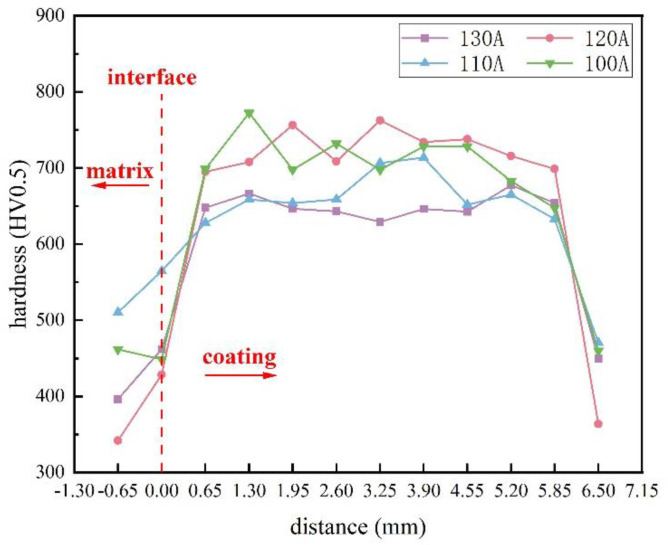
The relationship between the hardness of the welding layer, welding current, and position.

**Figure 19 materials-16-05059-f019:**
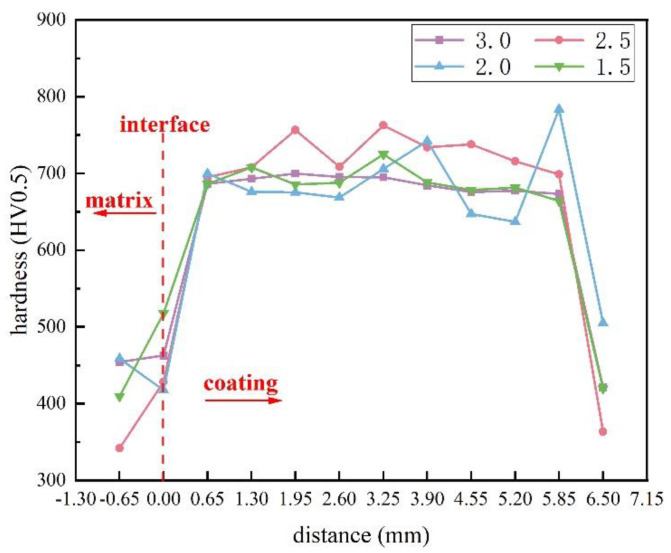
The relationship between the hardness of the welding layer, ion gas flow rate, and position.

**Table 1 materials-16-05059-t001:** The chemical composition of the matrix 42CrMo steel.

Elements	C	Mn	Mo	Cr	Ni	Fe
Content (wt.%)	0.45	0.60	0.20	1.00	0.25	Bal.

**Table 2 materials-16-05059-t002:** The chemical composition of the iron-based composite powder.

Elements	C	Mn	Mo	Ni	Si	Cr	V	Fe
Content (wt.%)	4.12	1.05	1.32	0.21	0.91	13.50	14.50	Bal.

**Table 3 materials-16-05059-t003:** PTA powder surfacing process parameters.

Number	Arc Current (A)	Ion Gas Flow Rate (L/min)	Protective Gas Flow Rate (L/min)	Powder Feeding Gas Flow Rate (L/min)	Powder Feed Rate (g/min)	Speed (rpm)
1	130	2.5	1.0	2.0	65	1.0
2	120					
3	110					
4	100					
5	120	3.0				
6		2.0				
7		1.5				

**Table 4 materials-16-05059-t004:** EDS elemental content of the Fe-based powder cross-section.

	Black A	Dark Gray B	Light Gray C
Element	wt.%	wt.%	wt.%
C	16.62	10.40	10.61
V	41.60	8.28	9.49
Cr	10.18	18.73	13.62
Mn	0.43	1.32	1.18
Fe	28.91	58.49	62.59
Si	0.44	0.91	1.16
Ni	0.22	0.19	0.28
Mo	1.60	1.68	1.07

**Table 6 materials-16-05059-t006:** Interface width of fusion zone under different arc currents.

	Point	1	2	3	4	5	6	7	8	Average Value
Arc Current										
130 A		10.66	11.48	9.29	12.84	10.38	12.02	10.93	10.38	11.00
120 A		7.92	9.56	7.92	11.75	11.48	11.20	12.02	10.36	10.28
110 A		6.63	5.52	4.97	6.35	7.10	6.08	9.39	8.29	6.79
100 A		7.09	5.26	5.67	6.48	5.06	6.28	6.48	5.55	5.98

**Table 7 materials-16-05059-t007:** EDS elemental content on the surface of the interface (the arc current is 110 A, and the ion gas flow rate is 2.5 L/min).

	A	B	C
Element	wt.%	wt.%	wt.%
C	0.03	0.01	0.02
V	83.72	24.15	13.51
Cr	8.72	10.41	31.70
Mn	0.15	0.98	1.40
Fe	5.21	62.22	50.26
Si	0.14	0.78	0.33
Ni	0.08	0.24	0.16
Mo	1.96	1.20	2.61

## Data Availability

The data that support the findings of this study are available on request from the corresponding author, [Cheng H.], upon reasonable request.

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
