# Peer review of "Composite Fe-Cr-V-C Coatings Prepared by Plasma Transferred-Arc Powder Surfacing"

_materials, 2023, doi:10.3390/ma16145059_

Round 1
Reviewer 1 Report
The following comments needs to be addressed in the revised manuscript by the authors:
1. Why the substrate was chosen as a channel (Fig.1) instead of plate? Please quote reasons in the revised manuscript.
2. What is the meaning of ion gas flow rate? What is the difference between ion gas flow rate and protective gas flow rate?
3. Is there any standard followed for the tensile testing of coated sample?
4. How the ion gas flow rate (Fig.13) influences the tensile strength? Explain with reasons.
5. What is the meaning of reticulation with reference to microstructure morphology?
Author Response
Response to Reviewer 1 Comments
Point 1: Why the substrate was chosen as a channel (Fig.1) instead of plate? Please quote reasons in the revised manuscript.
Response 1: Since the bond strength will be characterized by Tensile testing (it is necessary to ensure that the parallel section of tensile sample is just the coating without substrate, as shown in Fig. 2), in order to facilitate wire cutting sampling, the channel is used for surfacing.
Added explanation in the text above Fig. 1.
Point 2: What is the meaning of ion gas flow rate? What is the difference between ion gas flow rate and protective gas flow rate?
Response 2: Ionic gas is a key factor in controlling arc penetration, playing a role in igniting and compressing the arc. When other conditions are fixed, the ion gas flow rate increases, and the plasma flow force and arc penetration ability increase. In order to form stable small holes, sufficient ion gas flow rate is necessary. However, if the ion gas flow rate is too small, it may cause double arcs and damage the stability of the plasma arc.
The protective gas plays a role in protecting the stable combustion of the arc.
Ionic gas and protective gas can both be argon gas, which is introduced from two separate air intakes. The size of the ion gas and protective gas should be in an appropriate proportion, otherwise it will cause airflow turbulence, affect arc stability and protective effect.
Point 3: Is there any standard followed for the tensile testing of coated sample?
Response 3: Referring to the method of GB/T8642-2002 ‘Determination of tensile strength of thermal spraying’ innovative coating bonding strength means of detection. None of the breaks in the tensile test results in the paper are at the bond between the substrate and the coating, which can indicate that the bond strength is greater than the measured tensile strength, which is a semi-quantitative characterization.
Point 4: How the ion gas flow rate (Fig.13) influences the tensile strength? Explain with reasons.
Response 4: Added to the end of the first paragraph of 3.2 Tensile stress.
Point 5: What is the meaning of reticulation with reference to microstructure morphology?
Response 5: The composition of the mesh and the reasons for its formation are explained in the text at the top of Fig. 5. That is ‘Inter-dendritic M7C3 eutectic carbides are distrib-uted at the grain boundaries in the intermittent network. The reason for the reticula-tion of the hard phase of M7C3 is that the content of Cr is relatively low, so it cannot di-rectly precipitate the M7C3 phase’. The microhardness of this continuous network distributed eutectic carbide M7C3 is about HV1200, which can effectively enhance the hardness of the coating. The more evenly distributed the network structure, the better the uniform strengthening effect (mentioned in 3.3 Hardness).

Reviewer 2 Report
The paper will deserve publication after MINOR corrections as follows.
1. Chemical composition of the substrate steel should be indicated and, if possible, a rough profile of main alloying elements across the interfacial layer (could be determined by EDS). Besides, a way to draw the red borders of this layer (Fig. 8) wants explanations.
2. The text before Table 1 is confusing. On the one hand, the authors relate with Fig. 2 the POWDER MORPHOLOGY; on the other hand, the same figure is later (subsection 2.4) attributed to the TENSILE SPECIMEN scheme.
3. It is desirable to directly indicate thickness of tensile specimens (presumably 5 mm) to get sure that the loaded interface did not propagate into the substrate metal. Otherwise, the bonding assessment is questionable.
4. Fig. 9 displays the V-rich carbides (Type A) inside the COATING only, however the authors refer to this figure when discussing specific carbide compositions in the ITERAFCIAL layer. Please clear up!
5. There is a misprint (an excessive ‘not’) in item 2 of Conclusions.
Author Response
Response to Reviewer 2 Comments
Point 1: Chemical composition of the substrate steel should be indicated and, if possible, a rough profile of main alloying elements across the interfacial layer (could be determined by EDS). Besides, a way to draw the red borders of this layer (Fig. 8) wants explanations.
Response 1: The chemical composition of the matrix steel has been added as Table 1 and corresponding explanations have been provided in the text.
The approximate contour of the main alloy elements on the interface layer is represented by the EDS point scan in Fig. 9, the EDS line scan in Fig. 10, and the EDS surface scan in Fig. 11.
The method of drawing red boundaries is defined based on the organization of the heat affected zone and the significant differences within the weld overlay. Add to the text paragraph before Fig. 8.
Point 2: The text before Table 1 is confusing. On the one hand, the authors relate with Fig. 2 the POWDER MORPHOLOGY; on the other hand, the same figure is later (subsection 2.4) attributed to the TENSILE SPECIMEN scheme.
Response 2: The picture here is incorrectly numbered, it should be Fig. 3 not Fig. 2, it has been corrected. Fig. 2 is related to the tensile specimen program (subsection 2.4).
Point 3: It is desirable to directly indicate thickness of tensile specimens (presumably 5 mm) to get sure that the loaded interface did not propagate into the substrate metal. Otherwise, the bonding assessment is questionable.
Response 3: Has been supplemented in the text above Fig. 2 tensile specimen thickness of 1.5 mm. coatings thickness of about 5 mm, wire cutting 2 to 3 layers of sampling to do tensile to take the average value.
Point 4: Fig. 9 displays the V-rich carbides (Type A) inside the COATING only, however the authors refer to this figure when discussing specific carbide compositions in the ITERAFCIAL layer. Please clear up!
Response 4: The paragraph before Fig. 9 has been modified. The discussion was divided into two parts, namely the carbides in the interface layer (points B in Tables 5 and 7), the area near the interface layer, and the area at the center of the weld overlay layer (points A and C in Tables 5 and 7).
Point 5: There is a misprint (an excessive ‘not’) in item 2 of Conclusions.
Response 5: An excessive ‘not’ have been deleted.

Round 2
Reviewer 1 Report
The authors suitable addressed the queries.